# Sparse Reasoning is Enough: Biological-Inspired Framework for Abnormal Event Detection with Large Pre-trained Models

## Abstract

Abnormal Event Detection (AED) plays a crucial role in real-world applications, including security surveillance, autonomous driving, and industrial monitoring. Recent advances in large pre-trained models have opened new opportunities for training-free AED by leveraging rich prior knowledge and reasoning capabilities learned during pre-training. However, current studies typically rely on dense frame-level inference to ensure abnormal event coverage, incurring high computational costs and latency. This raises a fundamental question: Is this dense reasoning truly necessary when deploying large pre-trained models in AED? To answer this, we propose **ReCoAED**, a new framework inspired by the human nervous system's reflex arc-conscious reasoning stream, enabling adaptive frame processing to reduce redundant computation. It consists of two core streams: i) **Re**flex reacting stream: a lightweight CLIP compares frame features with prototype prompts to form decision vectors, which queries a dynamic memory of prior cases, enabling the system to rapidly determine whether to respond immediately with the memory or escalate the frame for deeper reasoning. ii) **Co**nscious reasoning stream: a medium-scale (7B) vision-language model analyzes novel frames, generating its event descriptions and anomalous scores to continuously update the dynamic memory. Periodically, an LLM reviews accumulated descriptions to identify new abnormal events, refine prototypes, and correct errors to realize self-evolution. Our extensive experiments show that ReCoAED reaches state-of-the-art training-free performance in UCF-Crime/XD-Violence datasets while reasoning on only **28.55%/16.04%** of frames used by the previous methods, showing that sparse reasoning is enough for effective large-model-based AED.

## 1 Introduction

Abnormal Event Detection (AED) aims to automatically identify anomalous events in video streams that exhibit deviation from normal patterns. It has attracted substantial attention due to its wide-range applications, including security surveillance Liu et al. (2018); Sultani et al. (2018), autonomous driving Yao et al. (2020), and industrial monitoring Pang et al. (2021), etc. Due to limited data and computational resources in the edge devices (e.g., camera, vehicle), it's crucial to develop efficient and generalizable AED systems to enable in-time responses to diverse abnormal events.

While conventional AED models exhibit high efficiency in limited conditions, they suffer from poor generalization and remain fragile in dynamic real-world deployments Zhu et al. (2022); Liu et al. (2023); Zanella et al. (2024); Wu et al. (2024). To overcome this, recent worksZanella et al. (2024); Yang et al. (2024); Kim et al. (2023); Tang et al. (2024); Du et al. (2024) explore the integration of Large Visual Language Model (LVLM) and Large Language Model (LLM), leveraging their broad prior knowledge from the large-scale pre-training. This emerging paradigm involves using i) LVLMs to produce rich textual descriptions of video frames and ii) LLMs to infer anomalous events from these descriptions Zanella et al. (2024). By decoupling abnormal event detection from low-level visual cues, this system achieves significantly improved generalization.

Despite these advantages, large-model AED systems face two critical deployment challenges: i) High computational cost of inference Samsi et al. (2023); Zhou et al. (2024), and ii) Dense frame-level analysis to ensure abnormal event coverage. These challenges inevitably compound the overall

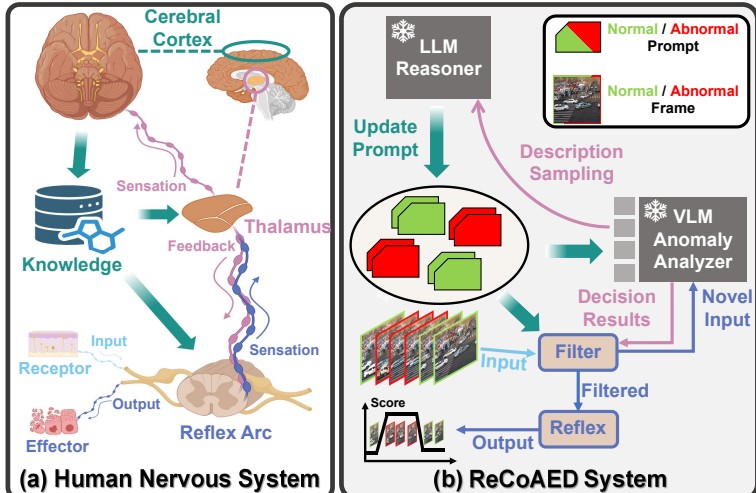

Figure 1: (a) The nervous system uses low-cost reflex arcs to process familiar signals, reserving cortical resources for unfamiliar ones through the thalamus–cortex loop. Bidirectional feedback enables top-down modulation of reflexes and bottom-up filtering of redundant information. (b) ReCoAED follows this architecture: the reflex component combines historic knowledge to filter trivial frames and spot novel ones. Novel frames then trigger conscious reasoning using a VLM for deeper analysis. An LLM closes the feedback loop by periodically refining both memory and prompts, incrementally improving the system.

computational burden. This raises a fundamental question: When leveraging powerful foundation models, is frequent inference across densely sampled frames truly necessary? While prior studies have validated the benefits of dense coverage in traditional AED pipelines Zanella et al. (2024); Yang et al. (2024); Wu et al. (2024), its utility in large-model systems remains underexplored and potentially suboptimal.

To address this challenge, we draw inspiration from how the human nervous system reduces the brain's burden through hierarchical processing Meunier (2009), allocating cognitive resources adaptively. As shown in Figure 1(a), it employs two complementary processing streams: i) *Reflex arc* that rapidly filters and responds to familiar, redundant signals and ii) *Conscious thinking* where the thalamus performs primary analysis Dehghani & Wimmer (2019) and cortical regions engage in further reasoning only on novel/complex signalsNani et al. (2019). These streams interact through a bidirectional feedback loopZagha (2020): the conscious reasoning influences the reflex arc's responses by manipulating memory and knowledge, while the reflex arc filters out trivial signals, thereby reducing the cognitive load on conscious thinking. This paradigm suggests that dense/uniform reasoning of the large pretrained models is not always necessary, inspiring us to develop a more efficient information routing framework for AED.

To this end, we propose **ReCoAED**, a novel framework that constructs the loop of efficient "**Re**flex" reacting with deeper "**Co**nscious" reasoning in Figure 1(b). The reflex reacting leverages a lightweight CLIP Radford et al. (2021) model to fuse visual features with prompts derived from textual event prototypes, producing a decision vector to query a dynamic memory of representative frames' records. If a frame falls within the coverage of the memory, it is considered trivial, and its anomalous score is retrieved directly, emulating low-latency, reflexive reacting. Frames that fail this check will be regarded as novel frames and activate the conscious reasoning stream, where a medium-scale (7B) VLM generates textual event descriptions and anomalous scores for these frames under the guidance of the textual event prototypes. These outputs are assembled as a new record and fed back to the reflex reacting stream's memory, refining the system's internal representations. To complete the feedback loop, the conscious reasoning also includes an LLM-based reasoner that periodically revisits generated descriptions to identify novel abnormal event types, revise earlier decisions, and adapt the prototype prompts, thus realizing a top-down self-evolution coupled with bottom-up filtering.

Therefore, our contributions are summarized as follows:

(1) To our best knowledge, we are the first to propose a biological-inspired AED framework that simulates the reflex arc-conscious reasoning of the human nervous system, significantly

reducing the computational burden of deploying VLM/LLM, improving processing speed while maintaining detection performance.

(2) In this framework, we establish a new closed-loop architecture coupling bottom-up filtering with top-down refinement: a lightweight reflex stream rapidly reacts to trivial frames via memory querying, while a VLM-LLM conscious reasoning stream analyzes only novel frames and progressively refines memory and prompts, enabling a self-evolving AED system.

(3) Through our extensive experiments, we demonstrate that dense-frame inference is unnecessary for large-model-based AED. Our ReCoAED achieves state-of-the-art, training-free performance on the UCF-Crime and XD-Violence datasets, while using only **28.55%** and **16.04%** of the frames compared to previous large-model-based approaches.

## 2 RELATED WORKS

### 2.1 ABNORMAL EVENT DETECTION

Abnormal Event Detection (AED) methods are commonly grouped into one-class, unsupervised, and weakly supervised settings, based on the availability of anomalies and labels during training. One-class AED assumes access to only normal videos and models normal patterns via direct statistical modeling Benezeth et al. (2009); Cheng et al. (2015); Hirschorn & Avidan (2023) or implicitly through proxy tasks Hirschorn & Avidan (2023); Liu et al. (2018); Xu et al. (2019); Shi et al. (2015); Singh & Pankajakshan (2018); Liu et al. (2022); Park et al. (2020); Li et al. (2020); Tao et al. (2024); Fang et al. (2020), including future frame prediction Liu et al. (2018); Xu et al. (2019), reconstruction Shi et al. (2015); Park et al. (2020), and contrastive learning Li et al. (2020); Tao et al. (2024). Unsupervised AED, on the other hand, assumes a mixture of normal and abnormal videos without labels Thakare et al. (2023b;a); Tur et al. (2023), often relying on the assumption that normal events dominate and applying similar modeling of normal data in one-class methods. Methods, *e.g.*GCLZaheer et al. (2022), further enhance the normal/abnormal distinction with cooperative training that uses pseudo-labels from autoencoder reconstruction errors to guide discriminators. Despite their scalability, these methods are prone to high false positives, as rare but normal events are often misclassified as anomalies Zhou et al. (2023); Lv et al. (2023), especially in open-world settings with diverse scenes. Weakly supervised AED utilizes video-level anomaly labels, typically via Multiple Instance Learning (MIL) Sultani et al. (2018). Many approaches enhance MIL with feature discrimination Tian et al. (2021); Chen et al. (2023) or self-training Zhong et al. (2019); Li et al. (2022); Shi et al. (2023), yet they remain limited by their dependence on training-time abnormal event types, which often differ from those encountered at test time. In summary, traditional methods rely heavily on distributional assumptions or supervision, making them brittle when faced with semantically diverse or unseen anomalies.

### 2.2 LVLM/LLM IN ABNORMAL EVENT DETECTION

To address these limitations, recent works integrate large-scale pre-trained LVLMs and LLMs into the AED Zanella et al. (2024); Yang et al. (2024); Kim et al. (2023); Tang et al. (2024); Du et al. (2024). These models possess broad prior knowledge and the reasoning capability acquired from large-scale pretraining and demonstrate strong performance. LAVAD Zanella et al. (2024) developed a training-free AED that predicts anomalies directly without training data. The method first utilizes the VLM to generate textual descriptions for video frames. Then, LLMs are prompted to analyze and give anomalous scores based on these descriptions. The AnomalyRuler Yang et al. (2024) further involves few-shot training videos to induce rules for the LLM to deduce anomalies during testing. With its strong capabilities, recent works have also developed new tasks in AED using LVLM/LLM, such as abnormal event Q&A in HAWK Tang et al. (2024) and cause analysis in CUVA Du et al. (2024). However, these new paradigms also introduce challenges in high inference cost and efficiency gaps that our proposed method seeks to address.

## 3 METHODOLOGY

### 3.1 PROBLEM SETUP

In this paper, we study the challenging task of training-free Abnormal Event Detection (AED). Given a test video, the objective of AED is to assign an anomalous score $s \in [0, 1]$ to each frame, indicating

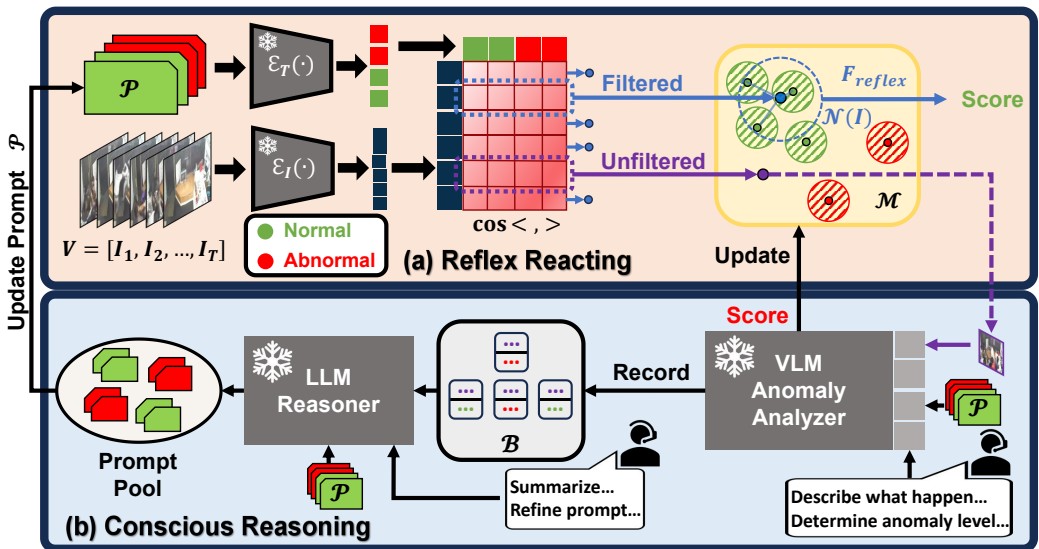

Figure 2: **ReCoAED** consists of **Re**flex reacting and **Co**nscious reasoning stream. The **Reflex reacting** stream employs a lightweight CLIP model to construct decision vectors $X_I$ by matching frame $I$'s visual features with textual event prototypes in the knowledge prompt $\mathcal{P}$. It then queries a dynamic core-set based memory $\mathcal{M}$ to decide if deep analysis is needed. If not, the anomalous score is retrieved directly via the reflex function through $I$'s neighbors in $\mathcal{M}$. Otherwise, the **Conscious reasoning** stream is activated to process $I$ with a VLM, generating event descriptions and anomalous scores under the guidance of $\mathcal{P}$, updating $\mathcal{M}$ with new records and contributing to the description set $\mathcal{B}$. Periodically, an LLM-based reasoner samples from $\mathcal{B}$ to revise prior decisions and refine $\mathcal{P}$, which in turn guides both streams for top-down refinement.

the probability of abnormal behavior. Unlike conventional approaches that rely on learning from the training set, training-free AED Zanella et al. (2024) requires detecting abnormal events in test videos without any access to training data. Formally, let a test video be represented as a sequence of $T$ frames $V = \{I_1, I_2, ..., I_T\}$. To perform inference, recent methods leverage a large vision-language model (VLM) $\Phi_{VLM}$ and a large language model (LLM) $\Phi_{LLM}$ to compute the anomalous scores $p$ according to the following standard pipeline:

$$C_I = \Phi_{VLM}(I), s = \Phi_{LLM}(C_I), I \in V, \tag{1}$$

where $\Phi_{VLM}$ generates the caption $C_I$ to describe the events in the frame $I$, and $\Phi_{LLM}$ reasons on $C_I$ to determine whether these events are anomalous. This approach of invoking large models on dense video frame sequences incurs high computational costs, limiting VLM/LLM-based methods in the real-world AED application. In this paper, we will show that this hard labor of large pretrained models is unnecessary by introducing the **ReCoAED** framework, which simulates how the human nervous system allocates cognitive resources adaptively.

### 3.2 INITIALIZATION

The overall framework of the ReCoAED is illustrated in Figure 2. It adopts an architecture that comprises two complementary streams: the reflex reacting stream and the conscious reasoning stream. The information transition between the two streams is completed through the knowledge prompt $\mathcal{P}$, a list of $L$ protocol event descriptions formulated as *person does something in some place* ($L$ is the total number of descriptions in $\mathcal{P}$). Although it is updated through conscious reasoning, an initialization is required. The initial $\mathcal{P}$ is set to contain 3 brief descriptions for both the normal/abnormal events in daily lives. The number of descriptions is updated into $L$ in the first round of LLM reasoning in Section 3.4. Due to the length limit, we put the full initiation of $\mathcal{P}$ in the Appendix B.

### 3.3 REFLEX REACTING

The reflex reacting draws inspiration from the human's reflex arc to filter out frames that can be addressed with the previous records with function $F_{filter}$, thereby enabling direct abnormal event detection using the function $F_{reflex}$ while uploading frames that require deep processing.

To build the foundation for $F_{filter}$ and $F_{reflex}$, we simulate how human reflexes are shaped by both innate perception and summarized knowledge, constructing a decision space by combining input frame features with the knowledge prompt $\mathcal{P}$ (derived from initialization and LLM reasoning) in the aligned vision-language feature space of a light-weight CLIP model. Specifically, for the frame $I$ in the test frame sequence $V$, we use the visual encoder $\mathcal{E}_I(\cdot)$ of CLIP to extract its visual representation $\mathcal{E}_I(I)$. Meanwhile, we also formulate $\mathcal{P}$ as a set of normal/abnormal event prototype descriptions:$\{p_i\}_{i=1}^L$. After processing through CLIP's text encoder $\mathcal{E}_T(\cdot)$, we obtain the corresponding text representations: $\{\mathcal{E}_T(p_i)\}_{i=1}^L$. The decision vector $X_I \in \mathbb{R}^L$ is then produced through the cosine similarity between $\mathcal{E}_I(I)$ and $\{\mathcal{E}_T(p_i)\}_{i=1}^L$:

$$X_I = [\gamma \langle \mathcal{E}_I(I), \mathcal{E}_T(p_1) \rangle, \gamma \langle \mathcal{E}_I(I), \mathcal{E}_T(p_2) \rangle, ..., \gamma \langle \mathcal{E}_I(I), \mathcal{E}_T(p_L) \rangle]^T, \tag{2}$$

where $\langle \cdot, \cdot \rangle$ represents this cosine similarity and $\gamma$ is CLIP model's log scale factor. The decision vector helps reduce task-irrelevant information. Since pre-trained visual encoders often lack task-specific priors and extract unrelated elements, computing similarities with event prototypes allows us to localize the fame within the task space defined by $\mathcal{P}$, effectively denoise distractions.

After obtaining the decision vector $X_I$, we can describe the two core functions as follows:

**Filter Function $F_{filter}$:** The function $F_{filter}$ identifies whether the input frame $I$ requires deep analysis by determining if it falls within the coverage of a dynamic memory $\mathcal{M}$ storing the representative prediction records. Each record is formulated as a dictionary-like data frame to describe an input-detection pair:

$$r = \{"visual" : \mathcal{E}_I(I), "decision\ vector" : X_I, "s" : s\}, \tag{3}$$

where $s$ is the recorded anomalous score of the frame $I$ processed from the raw output from the conscious reasoning in Section 3.4.

To fulfill the frame filtering, the memory $\mathcal{M}$ should be able to (1) effectively cover the main distribution of the frame stream with only a limited amount of frame records to reduce the request for large model inference; (2) establish a metric to determine whether the input frame fits in its coverage. To construct such $\mathcal{M}$, we adapt the idea from the greedy sampling from the core set algorithm Guo et al. (2022); Yang et al. (2023a); Roth et al. (2022). For each new frame $I$ with its $X_I$, if $\mathcal{M}$ is empty, we add its corresponding record to $\mathcal{M}$ directly. Otherwise, we use $F_{filter}$ to decide the next move:

$$F_{filter}(I, \mathcal{M}) = \mathbb{1}_{\{\min_{r \in \mathcal{M}} D(X_I, r["decision\ vector"]) > \epsilon\}}, \tag{4}$$

$F_{filter}$ calculate $X_I$'s shortest distance to existing decision vectors in the memory $\mathcal{M}$ with metric $D(\cdot, \cdot)$ (e.g. Euclidean distance). If the distance exceeds a predefined threshold $\epsilon$, $F_{filter}$ flags 1 and the frame is considered as a novel sample. It is sent to the large pretrained models in the conscious reasoning for further analysis. The resulting anomalous score $s$ from the conscious reasoning is averaged with $K$-nearest neighbors to mitigate the noise in the large model's predictions. Finally, the score $s$, the decision vector $X_I$, and the visual feature of the frame $I$ are assembled as a new record $r$, added into memory $\mathcal{M}$. If $X_I$ falls within the coverage of the memory $\mathcal{M}$, $F_{filter}$ flags 0. The corresponding frame $I$ is directly processed by the reflex function $F_{reflex}$.

Through $F_{filter}$, each record in $\mathcal{M}$ defines a hypersphere of radius $\epsilon$ around its decision vector, covering a local region. This ensures that all previously seen frames are either selected into $\mathcal{M}$ or covered by at least one hypersphere. Thus, $\mathcal{M}$ achieves compact coverage of known data while enabling novelty detection based on coverage failure.

**Reflex Function $F_{reflex}$:** The reflex function $F_{reflex}$ computes the anomalous score of a video frame $I$ based on its decision vector $X_I$ and the memory $\mathcal{M}$. Since $F_{filter}$ has determined that the frame fits in the coverage of $\mathcal{M}$, its anomalous score is inferred from its neighbors in $\mathcal{M}$. In practice, we find it effective to consider neighbors within the decision hypersphere of the radius $a \cdot \epsilon$ around the decision vector $X_I$. Also, we observe that large models are sensitive to signals of abnormal events, leading to potential false positives. To improve robustness, we make the final decision more conservative: a frame is labeled as anomalous only if all neighbors within the $a \cdot \epsilon$ radius range are also labeled as anomalous. Accordingly, the final anomalous score is defined as the minimum score among neighbors of the frame in the memory $\mathcal{M}$:

$$F_{reflex}(I, \mathcal{M}) = \min_{r_j \in \mathcal{N}(I)} (r_j["s"]), \tag{5}$$

$$\mathcal{N}(I) = \{r_j \in \mathcal{M} \mid \|X_I - r_j["decision\ vector"]\|_2 < a \cdot \epsilon\}, \tag{6}$$

where $\mathcal{N}(I)$ denotes the neighbor records of $I$ that falls within the decision hypersphere, and $r_j["s"]$ is the recorded anomalous score in $r_j$. The score from $F_{reflex}$ is then finalized as the output with a window smoothing, where the score is averaged together with scores of the $C$ temporal nearest frames to improve the prediction consistency.

### 3.4 CONSCIOUS REASONING

The conscious reasoning stream leverages large models to process novel frames identified by the reflex reacting, mimicking complex human cognitive processes. It consists of two modules: (1) a VLM anomaly analyzer to describe the events and determine the anomalous score of the novel frames; (2) an LLM Reasoner that summarizes from the previous cases to refine the knowledge prompt $\mathcal{P}$.

**VLM Anomaly Analyzer**: In this module, leveraging the knowledge prompt $\mathcal{P}$ and structured instructions, we integrate event description and abnormal event detection into one medium-scale(7B) VLM $\Phi_{VLM}$ to analyze the frame step-by-step. Given a video frame $I$ and the knowledge prompt $\mathcal{P}$, we construct the instruction prompt $P_{VLM}$ to guide the VLM anomaly analyzer to: (1) first describe events in the frame, (2) then compare them with prototypes in $\mathcal{P}$ and then (3) assign an anomalous score chosen from the option list: *OPTIONS*. The *OPTIONS* contains 9 distinct anomalous scores (real numbers from 0 to 1). Each score in *OPTIONS* is compiled with its explanation based on the match/mismatch degree between the event and the normal/abnormal prototypes, thereby reducing the arbitrary decisions of VLM $\Phi_{VLM}$. Finally, $P_{VLM}$ specifies output format as $(des, s)$ where $des$ is the detailed description of the events in the frame and $s$ is the anomalous score to formulate a new record for memory $\mathcal{M}$. Therefore, the prompt for the VLM $\Phi_{VLM}$ is the concatenation of $P_{VLM}$ and $\mathcal{P}$: $P_{VLM} \circ \mathcal{P}$. The details of $P_{VLM}$ and *OPTIONS* can be found in the Appendix C. The output in this stream is as follows:

$$(des, s) = \Phi_{VLM}(I, P_{VLM} \circ \mathcal{P}) \tag{7}$$

**LLM Reasoner:** At a fixed interval of $N$ videos, the LLM analyzes accumulated $(des, s)$ pairs stored in a temporal description set $\mathcal{B}$ to emulate human-like conscious summarization, enabling refinement of $\mathcal{P}$ and thereby enhancing both $\Phi_{VLM}$'s abnormal event detection and the construction of decision vectors in the reflex reacting stream. Specifically, we randomly sample a subset $\mathcal{B}' \subseteq \mathcal{B}$ of $b$ pairs, and design a prompt $P_{LLM}$ asking the LLM to clarify the previous $\mathcal{P}$ to describe the newly discovered normal/abnormal events better, so that $\mathcal{P}$ can better represent the global data. The details of the $P_{LLM}$ can be found in the Appendix D. Therefore, the new knowledge prompt $\mathcal{P}$ can be updated with the concatenation of $P_{LLM}$, $\mathcal{P}$ and $\mathcal{B}'$:

$$\mathcal{P} \leftarrow \Phi_{LLM}(P_{LLM} \circ \mathcal{P} \circ \mathcal{B}'). \tag{8}$$

Notably, we update the memory $\mathcal{M}$ to fit the change, where we recalculate the decision vector of every $r \in \mathcal{M}$ with recorded visual features and $\{p_i\}_{i=1}^{L}$ formulated from updated $\mathcal{P}$ using Eq. 2. Also, we re-evaluate the anomalous scores with $F_{reflex}$ for the historic frames with the new memory $\mathcal{M}$ to correct previous mistakes, improving the overall accuracy. After the feedback, $\mathcal{B}$ is emptied for the next round to ensure the LLM always summarizes from the new events.

In summary, the framework forms a top-down self-improvement coupled with bottom-up filtering, demonstrated as Algorithm 1.

---

**Algorithm 1: ReCoAED Framework**

**Input:** Test Video set $\mathcal{V}$, image/text encoder $\mathcal{E}_I(\cdot)/\mathcal{E}_T(\cdot)$, large pretrained models $\Phi_{VLM}, \Phi_{LLM}$, initial prompt $\mathcal{P}$.
**Output:** Anomalous scores $s$ for each frame.
1: Initialize $\mathcal{M} = \emptyset$, $\mathcal{B} = \emptyset$, $n = 0$;
2: **for** each video $V$ in $\mathcal{V}$ **do**
3:   **for** each frame $I$ in $V$ **do**
4:     Formulate $\mathcal{P}$ as $\{p_i\}_{i=1}^{L}$ and calculate $X_I$ with Eq. 2;
5:     **if** $F_{filter}(I, \mathcal{M}) == 1$ **then**
6:       $(des, s) = \Phi_{VLM}(I, P_{VLM} \circ \mathcal{P})$;
7:       $r \leftarrow \{visual : \mathcal{E}_I(I), decision\ vector : X_I, s : s\}$;
8:       Add $r$ to $\mathcal{M}$; add $(des, s)$ to $\mathcal{B}$;
9:     **else**
10:       $s \leftarrow F_{reflex}(I, \mathcal{M})$;
11:     **end if**
12:   **end for**
13:   $n \leftarrow n + 1$;
14:   **if** $n \bmod N == 0$ **then**
15:     Sample $b$ records from $\mathcal{B}$ to form the subset $\mathcal{B}'$;
16:     Update $\mathcal{P} \leftarrow \Phi_{LLM}(P_{LLM} \circ \mathcal{P} \circ \mathcal{B}')$;
17:     Update $X_I$ for $\forall r \in \mathcal{M}$ with new $\mathcal{P}$ by Eq. 2;
18:     Re-evaluate $s$ for the previous frame using $F_{reflex}$.
19:     $\mathcal{B} \leftarrow \emptyset$;
20:   **end if**
21: **end for**

## 4 EXPERIMENTS

We evaluate our framework on UCF-Crime Sultani et al. (2018) and XD-Violence Wu et al. (2020), focusing on Training-free AED accuracy and the number of frames processed by large pretrained models. Unlike prior methods relying on dense frame inference, our approach achieves higher performance with only a sparse subset of the frame used by the previous methods, demonstrating strong efficiency and effectiveness.

### 4.1 EXPERIMENTAL SETUPS

**Dataset and Test Settings:** We evaluate our framework on the UCF-Crime Sultani et al. (2018) and XD-Violence Wu et al. (2020) datasets, which consist of real-world surveillance footage with diverse events. **UCF-Crime** includes 1,900 untrimmed videos covering 13 abnormal events. The training set has 800 normal and 810 abnormal videos, while the test set contains 140 normal and 150 abnormal videos with frame-level annotations. **XD-Violence** contains 4,754 multi-modal videos from movies and YouTube, spanning 6 types of violent events, with 3,954 training

Table 1: Comparison against one-class, unsupervised, and training-free methods on UCF-Crime dataset

| Method | Supervised mod | AUC(%) | Frames for VLM or LLM |
|---|---|---|---|
| BODS Wang & Cherian (2019) | one-class | 68.26 | - |
| GODS Wang & Cherian (2019) | one-class | 70.46 | - |
| GCL Zaheer et al. (2022) | unsupervised | 74.20 | - |
| DYANNET Zaheer et al. (2022) | unsupervised | 79.26 | - |
| ZS CLIP Zanella et al. (2024) | Training-free | 53.16 | - |
| Baseline(Qwen2.5-VL-7B) | Training-free | 79.22 | 69,344 |
| LAVAD Zanella et al. (2024) | Training-free | 80.28 | 69,344 |
| **ReCoAED** | Training-free | **82.28** | **19,797** |

Table 2: Comparison against one-class, unsupervised, and training-free methods on XD-Violence dataset

| Method | Supervised mod | AP(%) | AUC(%) | Frames for VLM or LLM |
|---|---|---|---|---|
| HASAN et al. Hasan et al. (2016) | one-class | - | 50.32 | |
| LU et al. Lu et al. (2013) | one-class | - | 53.56 | |
| BODS Wang & Cherian (2019) | one-class | - | 57.32 | - |
| GODS Wang & Cherian (2019) | one-class | - | 61.56 | - |
| RAREANOM Thakare et al. (2023a) | Unsupervised | - | 68.33 | - |
| ZS CLIP Zanella et al. (2024) | Training-free | 17.83 | 38.21 | - |
| Baseline(Qwen2.5-VL-7B) | Training-free | 55.21 | 83.59 | 145,649 |
| LAVAD Zanella et al. (2024) | Training-free | 62.01 | 85.36 | 145,649 |
| **ReCoAED** | Training-free | **65.66** | **86.38** | **23,362** |

and 800 test videos. We follow LAVAD Zanella et al. (2024)'s experimental setup, testing on the test sets without using training data or annotations. We follow the experimental setup of LAVAD Zanella et al. (2024), evaluating our framework on the test sets of both datasets without seeing or training with training videos or annotations. Notably, we randomly shuffle the test sets, which originally contain consecutive samples of the same event type. This shuffling increases task difficulty by preventing the model from exploiting event continuity, providing a more rigorous assessment of our framework's ability to generalize to unseen scenes and events, and better simulating real-world conditions.

**Performance Metrics:** Following standard practice, we report the frame-level ROC AUC on UCF-Crime and XD-Violence datasets, which is regarded as a fair metric for class-imbalanced tasks like abnormal event detection Sultani et al. (2018); Liu et al. (2018). Also, we evaluate the frame-level average precision (AP), i.e., the area under the precision-recall curve, on XD-Violence following the setting in the work Wu et al. (2020); Zhou et al. (2023)

**Implementation Details:** The basic video inputs are represented as frame sequences uniformly sampled from each video every 16 frames following LAVAD Zanella et al. (2024); Wu et al. (2024). The CLIP model in the reflex reacting is set as pretrained CLIP-ViT-B/16 Radford et al. (2021). The parameter $\epsilon$ in the memory $\mathcal{M}$ is set as 2.0 for UCF-Crime and 4.0 for XD-Violence to ensure the total compressed rate of the inputs falls in the reasonable range of $15\% - 30\%$. The neighbor number $K$, the window size $C$, and the radius coefficient $a$ in $F_{reflex}$ are set uniformly for both datasets as 16,4, and 2. In conscious reasoning, we implement the VLM as the Qwen2.5-VL-7B Bai et al. (2025) model that is widely used by today's open-source community. At last, we set LLM as the Deepseek-V3 Liu et al. (2024) model. The interval parameter $N$ is set as 10. The length $L$ for $\mathcal{P}$ that is summarized by the LLM is set to be 20, with a half-to-half split for normal and abnormal events. The parameter $b$ for the size of the subset $\mathcal{B}'$ is set as 90. In practice, we sample $b/2$ $(des, s)$ pairs with score $s > 0.5$ and $b/2$ pairs with $s < 0.5$ to ensure normal/abnormal balance in $\mathcal{B}'$. In the initialization round, when the total length of $\mathcal{P}$ is set as 3, we shrink $\epsilon$ as 1.2 uniformly at the beginning to ensure sufficient sampling for the LLM to analyze. After the first round of the LLM reasoning, $\epsilon$ is set to be the predefined value in each dataset. To further validate the effectiveness of our approach, we implement a **Baseline** also using Qwen2.5-VL-7B, which predicts only through the VLM anomaly analyzer $\Phi_{VLM}$ with $P_{VLM}$, without the reflex reacting or the LLM-based refinement.

### 4.2 EXPERIMENTAL ANALYSIS AND VISUALIZATION

**Performance Analysis:** The experimental results are presented in Table 1 and Table 2. In the tables, one-class methods are trained using only the normal videos from the training set, while unsupervised

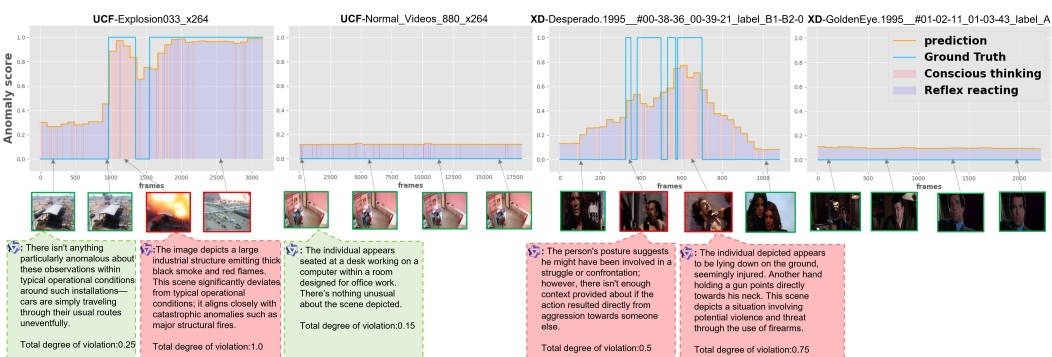

Figure 3: Visualization of the predictions made by the reflex reacting and the conscious reasoning.

Table 3: Efficiency comparison against previous frameworks and sampling strategies on UCF-Crime

| Methods | AUC(%) | Frames for VLMs/LLMs | FPS$_{sampled}$ | FPS$_{overall}$ |
|---|---|---|---|---|
| LAVAD | 80.28 | 69,344 | 0.49 | 7.86 |
| Baseline(uniform) | 79.22 | 69,344 | 2.78 | 44.57 |
| + MGSampler | 77.49 | 34,372 | 5.72 | 91.71 |
| + Segment-Sampler | 75.90 | 22,725 | **8.49** | **136.13** |
| ReCoAED (Ours) | **82.28** | 19,797 | 5.76 | 92.40 |

Table 4: Runtime of ReCoAED on UCF-Crime

| Component | Runtime(s) |
|---|---|
| Reflex reacting | 657.6 |
| VLM Analyzer | 10,451.61 |
| LLM Reasoner | 923.93 |
| Total | 12,033.14 |

methods use the entire training set without access to any annotations. Training-free methods do not utilize any training data and perform inference directly on the test set. Notably, the "ZS CLIP" refers to a zero-shot CLIP-based approach that computes anomalous scores by contrasting prompts for normal/abnormal events, following the same prompt design used in LAVADZanella et al. (2024).

As shown in Table 1 and Table 2, our method significantly outperforms traditional one-class and unsupervised abnormal event detection approaches. Most importantly, our framework achieves notable improvements in both accuracy and efficiency compared to previous training-free methods based on large pretrained models. Specifically, compared with the state-of-the-art training-free method LAVAD Zanella et al. (2024), our framework achieves a 2.00% AUC improvement on UCF-Crime, a 3.65% AP gain, and a 1.02% AUC gain on XD-Violence. In addition to the superior performance, our method significantly reduces the computational burden on large pretrained models. Our approach processes only 28.55% and 16.04% of the frames used by LAVAD in UCF-Crime and XD-Violence with large pre-trained models. Also, compared to the baseline using Qwen2.5-VL-7B, our method achieves a 3.06% AUC improvement on UCF-Crime and gains of 10.45% in AP together with 2.79% in AUC on XD-Violence, while also achieving computational savings on the large pre-trained models. These results confirm that our framework delivers both superior detection performance and a low computational burden for large pre-trained models.

**Efficiency Analysis:** To assess further efficiency, we evaluate the processing speed of ReCoAED against LAVAD and two traditional sampling schemes on UCF-Crime: MGSamplerZhi et al. (2021)(motion-based keyframe) and Segment SamplerYang et al. (2023b)(fixed-position frames) with a window smoothingLv et al. (2023) (Table 3) on two Nvidia A100 GPUs. We report FPS$_{sampled}$, which is computed from the number of frames from the basic uniform sampling (69344 frames) divided by the total runtime, and FPS$_{overall}$ calculated using the total number of original video frames (1111808 frames) divided by the total runtime. As is shown in Table 3, on the one hand, ReCoAED achieves a significantly higher processing speed than the previous method, LAVAD. On the other hand, although previous sampling schemes boost the baseline throughput, they cause notable performance drops. In contrast, using a similar number of frames compared with previous sampling methods, ReCoAED attains a 107% speedup over the baseline and improves the performance. Table 4 shows that the periodically invoked LLM (called through API) contributes only 7.68% of total inference time. The main gains come from the Reflex reacting, which filters 71.45% of frames before the VLM Anomaly Analyzer and scores them far faster than the VLM, substantially reducing the total runtime.

**Visualization**: We further validate the roles of both streams through prediction visualizations. As demonstrated in Figure 3, the reflex reacting (light purple regions) dominates the output scores, especially for routine/repetitive events in normal videos, effectively reducing reliance on the large model. For novel events like the explosion in `UCF-Explosion033_x264` case, the reflex reacting identifies unfamiliar patterns and activates the conscious reasoning for accurate description and detection. Once the abnormal event is recorded, similar events in future frames are handled directly

Table 5: The effectiveness of the conscious reasoning's components

| Feedback $\mathcal{P}$ to reflex | Feedback $\mathcal{P}$ to $\Phi_{VLM}$ | OPTIONS | AUC(%) |
|---|---|---|---|
| ✗ | ✗ | ✗ | 70.80 |
| ✗ | ✓ | ✓ | 74.62 |
| ✓ | ✗ | ✗ | 77.83 |
| ✓ | ✓ | ✗ | 79.92 |
| ✓ | ✓ | ✓ | 82.28 |

Table 6: The effectiveness of the reflex reacting's components

| $F_{reflex}$ | Use minimum among neighbors | Window smooth | AUC(%) |
|---|---|---|---|
| ✗ | ✗ | ✗ | 79.24 |
| ✗ | ✗ | ✓ | 80.36 |
| ✓ | ✗ | ✗ | 79.97 |
| ✓ | ✓ | ✗ | 80.93 |
| ✓ | ✓ | ✓ | 82.28 |

by the reflex reacting, minimizing repeated inference. This illustrates the coordination between streams: the reflex reacting offers responses for the most trivial inputs, while the conscious reasoning only handles novel cases, greatly reducing the computational burden of large models.

# 5 ABLATION STUDY

## 5.1 EFFECTIVENESS OF COMPONENTS

We evaluate the effectiveness of both streams in our framework through ablations. Specifically, we isolate the reflex reacting and the conscious reasoning, keeping one fixed while testing the other.

Table 5 presents results for the conscious reasoning stream. We examine whether the dynamically updated prompt $\mathcal{P}$ can improve overall performance, and assess the role of both the knowledge prompt and the anomaly-level option list *OPTIONS* in enhancing VLM $\Phi_{VLM}$'s detection. When $\mathcal{P}$ is blocked from both the reflex reacting and $\Phi_{VLM}$, performance drops significantly below the baseline. This is because $F_{filter}$ cannot construct decision vectors aligned with the current task space, leading to poor distance estimation in $F_{reflex}$ and ineffective frame selection. Additionally, when the $\Phi_{VLM}$ lacks updated knowledge, it also reduces detection accuracy. As shown in rows 2 and 3 of Table 5, enabling knowledge prompt $\mathcal{P}$ transmission to the VLM anomaly analyzer and reflex reacting improves performance by 7.03% and 3.82%, respectively. Rows 4 and 5 further evaluate the impact of *OPTIONS*. In Row 4, we set the VLM to produce a raw anomalous score in $[0, 1]$ as in LAVAD when *OPTIONS* is absent. In row 5, by incorporating *OPTIONS*, the framework yields a 1.35% gain, highlighting the benefit of providing interpretable anomaly cues.

We further examine the reflex reacting stream, focusing on $F_{reflex}$, as the efficiency of $F_{filter}$ is validated in Section 4.2. Table 6 compares 3 testings: (1) Without $F_{reflex}$, the model uses the prediction of the nearest neighbor in memory $\mathcal{M}$ as output, resulting in a 3.04% performance drop; (2) Rows 3 and 4 compare aggregation strategies in $F_{reflex}$: using the average vs. the minimum prediction score. The latter yields a 0.96% improvement, better reducing false positives; (3) Applying a temporal smoothing window improves temporal consistency, boosting accuracy by 1.35% and 1.12% with/without $F_{reflex}$, suggesting greater effectiveness when combined with $F_{reflex}$.

## 5.2 SENSITIVITY TO HYPER-PARAMETERS

We further analyze the sensitivity of our framework to hyperparameters. Here we take the radius $a \cdot \epsilon$ of the decision hypersphere, and the number of neighbors $K$ as examples. As illustrated in Figure 4, a small radius makes the model fragile to noisy entries, while a large one risks including irrelevant frames. Similarly, a small $K$ fails to suppress noise, whereas a large $K$ may bury the original score. Overall, performance remains stable (within 1.8% and 0.23%),

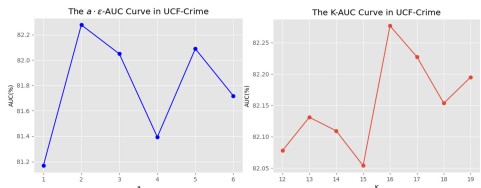

Figure 4: The ablation on parameter $K$ and the radius of the decision hypersphere

demonstrating strong robustness. More ablations on hyperparameters are included in the Appendix.

# 6 CONCLUSION AND LIMITATION

To summarize, in this paper, we demonstrate through our biological-inspired dual stream Re-CoAED framework and extensive experiments that the previous high-frequency reasoning with large pre-trained models in AED is unnecessary. ReCoAED reduces the computational burden of the VLM/LLM models by using a reflex reacting to filter and respond to trivial inputs, and leaves the large model to deal with novel inputs to refine the entire system. The experiments and visualization demonstrate that ReCoAED achieves state-of-the-art performance, with the reflex-reacting handling the majority of the inputs, thereby significantly reducing the computational burden of the large models. However, our work still has the limitation that the current framework is built on a limited set of models. In future work, we plan to extend our study to a broader range of VLM/VLM choices to further investigate the generalization ability of our framework.

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

## A    LLM Usage

Large Language Models (LLMs) were used to aid in the writing and polishing of the manuscript. LLM was not involved in the ideation, research methodology, or experimental design. All research concepts, ideas, and analyses were developed and conducted by the authors. The contributions of the LLM were solely focused on improving the linguistic quality of the paper, with no involvement in the scientific content or data analysis.

The authors take full responsibility for the content of the manuscript, including any text polished by the LLM. We have ensured that the LLM-generated text adheres to ethical guidelines and does not contribute to plagiarism or scientific misconduct.

## B    Initialization and Formation of the Knowledge Prompt $\mathcal{P}$

For the proposed framework, ReCoAED, the knowledge prompt $\mathcal{P}$ plays a pivotal role in enabling effective knowledge transfer. Apart from our description of $\mathcal{P}$ in Section 3.2 of the main text, here we provide more information about its initialization and formation.

Although $\mathcal{P}$ is primarily updated in this work via the LLM reasoner in the conscious reasoning by summarizing past cases, it is first initialized to provide the framework with essential prior knowledge. In this paper, we adopt the following unified prompt formulation to initialize $\mathcal{P}$ for both the UCF-Crime and XD-Violence datasets as in the Figure 5 (a). The initial prompt list is evenly divided into the normal group and the abnormal group to maintain the information balance.

Three simple and general descriptions of the normal events: *1. People normally walk, stand, or sit while doing daily things. 2. Cars drive on the road normally. 3. Normal scene without any event taking place.*

Three simple general descriptions of the abnormal events: *1. People are committing criminal activities. 2. People have strange postures that reflect possible crimes. 3. Accidents/disasters happen in the background.*

These initial event prototypes provide a general description of both normal and abnormal events without delving into specific behaviors/actions, details that are instead summarized and extracted by the framework. We can easily formulate such descriptions without requiring intricate knowledge of the particular scenes or events involved. While maintaining their generality and simplicity, these prototypes also initialize the structure of the framework's task space, offering a fundamental semantic partitioning between normal behaviors and abnormal events such as criminal activities and accidents. This, in turn, serves to guide subsequent large-scale models in the process of analyzing and summarizing the prototypes.

These prototypes are then rewritten as the knowledge prompt $\mathcal{P}$, a list of prompts formulated as *"An image contains: event prototype*, as shown in 5 (b), to enable the construction of the decision space in the reflex reacting. The knowledge prototypes in the prompt list above are concatenated in the form of the template in Figure 5 (c), as a code book for the VLM anomaly analyzer to refer to and make their descriptions and judgments of whether the image contains abnormal events. Finally, we require the LLM reasoner to output $L$ prototypes from the previous cases, following the same form as the initial descriptions. Also, to maintain the normal/abnormal balance in the prompts, we ask the LLM to output up to $L/2$ normal prototypes and $L/2$ abnormal prototypes to make sure the new abnormal knowledge is recorded in the sample-unbalanced abnormal event detection task.

## C    Formulation of the Prompt $P_{VLM}$

In this section, we discuss the formation of the prompt $P_{VLM}$. As described in Section 3.4, the prompt fulfills the following functions: (1) describe events in the frame; (2) compare them with prototypes in the knowledge prompt $\mathcal{P}$; (3) assign an anomalous score chosen from the option list: *OPTIONS*; (4) regulate the output format for future analysis. In order to instruct the VLM anomaly analyzer to achieve these functions, we design the $P_{VLM}$ as in Figure 6.

As is shown in the Figure 6 (a), the overall formation of the prompt $P_{VLM}$ instructs the VLM anomaly analyzer to describe all the events in the image in detail, compare them against the event

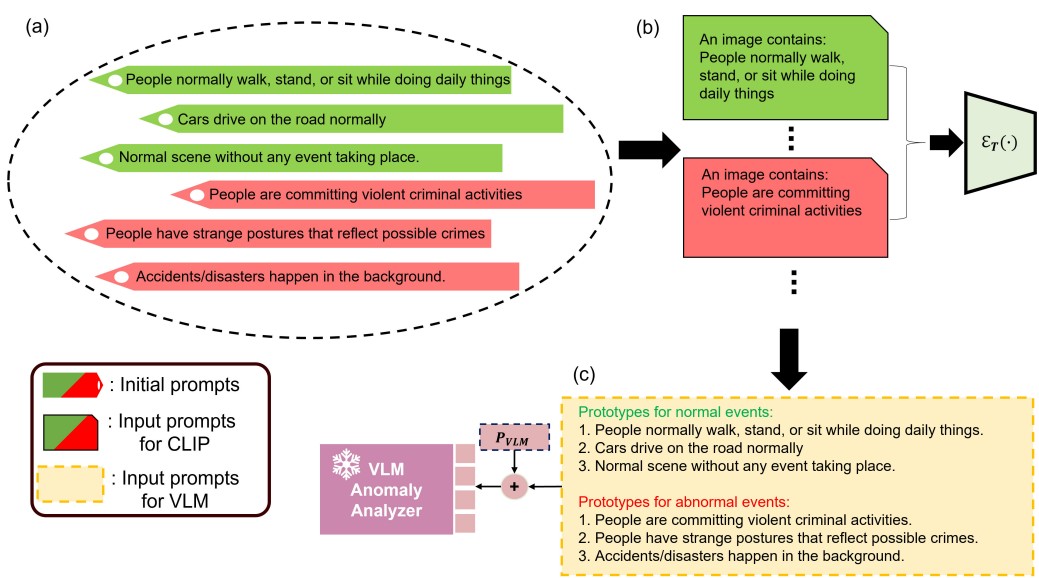

Figure 5: Initialization of the knowledge prompt $\mathcal{P}$.

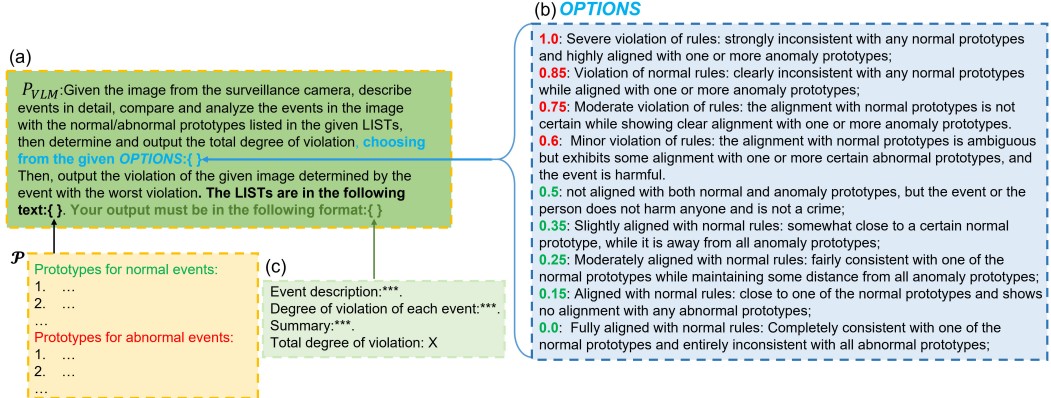

Figure 6: The design of the $P_{VLM}$ prompt for the VLM anomaly analyzer.

prototypes in the given knowledge prompt $\mathcal{P}$( in the code book form) and determine their anomalous score, which are chosen from the option list *OPTION* in the Figure 6 (b).

As demonstrated, *OPTION* constrains the scores the model can choose from and requires it to make decisions based on explicit interpretations of these scores, thereby reducing the possibility of subjective or hasty judgments. Each score explanation is derived from the degree of alignment between the observed event and the predefined prototypes of normal or abnormal events. The scores are determined based on the following principle: if an event shows alignment with one or more abnormal prototypes, it should be assigned a score higher than 0.5, with the exact score reflecting the strength of this alignment. Conversely, if the event does not match any abnormal prototype, it is assigned a low anomalous score, which is determined based on how well it aligns with normal prototypes; the stronger the alignment with normal behavior, the lower the anomalous score.

Further, $P_{VLM}$ asks the model to determine the degree of the event of the worst abnormal situation, as the image may contain normal events that can hinder the model's decision. At last, the VLM anomaly analyzer is required to output the information in the given format in the Figure 6 (c) to allow the extraction of the event descriptions and anomalous scores for the LLM and the reflex reacting.

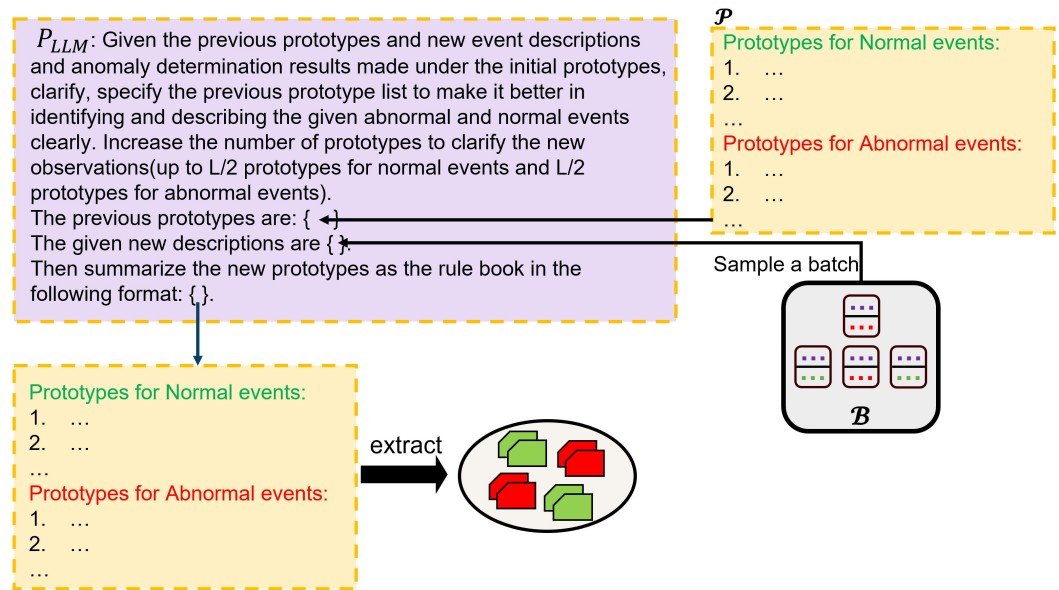

Figure 7: The design of the $P_{LLM}$ prompt for the LLM reasoner.

Notably, in the main text, we only demonstrate the **Summary** and **Total degree of violation** part of the VLM output for visualization in Figure 3 due to the length limit.

## D  FORMULATION OF THE PROMPT $P_{LLM}$

In this section, we further introduce the prompt $P_{LLM}$ used in the conscious reasoning. The primary function of this prompt is to instruct the model to leverage the accumulated historical anomaly descriptions/detection cases to revise and expand the previous knowledge prompt $\mathcal{P}$, finally generating an updated version. The detailed information about the prompt $P_{LLM}$ is shown in Figure 7.

As is demonstrated, $P_{LLM}$ gathers the previous cases sampled from the description set $\mathcal{B}$ and instructs the LLM to clarify and specify the previous prompts to fit the new cases. This process requires the model to analyze previously used event prototypes that may have led to inaccurate discrimination, and to concretize vague event descriptions by incorporating new event information. In doing so, the updated knowledge prompt can better support the lower-layer model in sample filtering and abnormal event discrimination. The outputs are new prototypes which are organized in the same form as the code book form of $\mathcal{P}$. These prototypes are extracted from the output and are treated as the new knowledge prompt, fed back to the reflex reacting and the VLM anomaly analyzer.

## E  ABLATIONS ON MORE HYPERPARAMETERS

In this section, we present additional ablation studies to further validate our framework's effectiveness. Specifically, the ablations focus on three key parameters: $\epsilon$, which determines the volume of the memory $\mathcal{M}$, the interval parameter $N$, which determines the frequency of LLM reasoning, and the parameter $L$, which controls the number of event prototypes maintained in the knowledge prompt $\mathcal{P}$.

We first investigate $\epsilon$, which determines the volume of the memory $\mathcal{M}$. As shown in Table 7, the choice of $\epsilon$ is crucial. When $\epsilon$ is too large, new frames are rarely added to the memory $\mathcal{M}$, which reduces the total number of recorded frames. This leads to an inaccurate representation of the distribution of seen frames, making it difficult to filter input frames effectively and ultimately degrading performance. In contrast, a small $\epsilon$ leads to excessive storage of trivial frames, increasing noise sensitivity, and triggering unnecessary large-model inference.

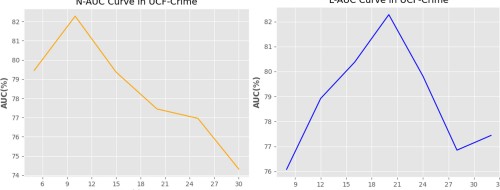

| $\epsilon$ | 1.6 | 1.8 | 2.0 | 2.2 | 2.4 | 2.6 |
|---|---|---|---|---|---|---|
| AUC(%) | 77.09 | 80.01 | 82.28 | 81.45 | 77.59 | 77.97 |
| frames for $\Phi_{VLM}$ | 30,309 | 24,337 | 19,797 | 18,246 | 13,338 | 10,725 |

Table 7: Ablation on the parameter $\epsilon$ in $F_{filter}$

Figure 8: The ablation on the parameter $N$ and the parameter $L$.

The parameter $N$ is a critical component of our framework, as it determines how frequently the LLM is invoked and, consequently, how often the memory $\mathcal{M}$ and detection results are refined. As shown in Figure 8, we report the results under different values of $N$. It is evident that overly frequent summarization does not improve model performance; in fact, it leads to a noticeable degradation. This can be attributed to the limited scope of prototype summarization when the LLM is invoked too frequently, which weakens the generalization ability of the event prototypes. Frequent updates to the decision space also introduce instability, further degrading performance.

Conversely, setting $N$ too large also results in significant performance drops. In such cases, as shown in Figure 8, the large volume of videos causes an accumulation of event descriptions, making novel and informative anomalies relatively sparse. As a result, the model struggles to extract meaningful new knowledge. Moreover, a long summarization interval delays the update of outdated

In addition, we evaluate the impact of the prototype count parameter $L$ on the performance of our framework. The results are presented in Figure 8. As shown, when $L$ is too small, the upper bound of knowledge that the framework can retain is limited, making it insufficient to capture the complexity of the dataset, especially for datasets like UCF-Crime and XD-Violence, which exhibit highly diverse and complex distributions. This leads to a drop in detection accuracy.

On the other hand, when $L$ is set too large, the knowledge prompt $\mathcal{P}$ becomes overly overpopulated with trivial information, which hinders the clear separation between different types of events within the decision space. Moreover, the excessive information may introduce unnecessary interference for the VLM anomaly analyzer when distinguishing between normal and abnormal events, ultimately resulting in degraded performance.

# F VISUALIZATION OF ReCoAED'S SELF-EVOLUTION VIA THE KNOWLEDGE PROMPT $\mathcal{P}$

In this section, we demonstrate the self-evolution capability of our framework, ReCoAED, through the visualization of knowledge prompt $\mathcal{P}$'s self-refinement. As discussed earlier, $\mathcal{P}$ is a crucial component of our framework, serving as the bridge between the reflex and conscious reasonings. They provide essential guidance to the VLM anomaly analyzer and play a key role in shaping the decision space of the reflex reacting. More specific and task-aligned knowledge prompt leads to better overall performance of the framework in abnormal event detection.

Figure 9 and Figure 10 illustrate the final knowledge prompt obtained after testing. For the sake of visual clarity, we formulate the knowledge prompt in the code book form. Remarkably, without any human intervention or supervision during the process, our framework can automatically extract and summarize the types of abnormal events present in the dataset through a combination of top-down refinement and bottom-up filtering. As indicated by the arrows connecting the abnormal event prototypes to the ground-truth abnormal event classes, most of the event prototypes in the knowledge prompt $\mathcal{P}$ exhibit a clear semantic alignment with one or more ground-truth abnormal event categories. This demonstrates that our framework can effectively identify key information about abnormal events through its self-evolving process. As a result, it is capable of autonomously adapting to different datasets, significantly enriching and grounding the initial knowledge prompt, and providing meaningful guidance for the models in the lower layers.

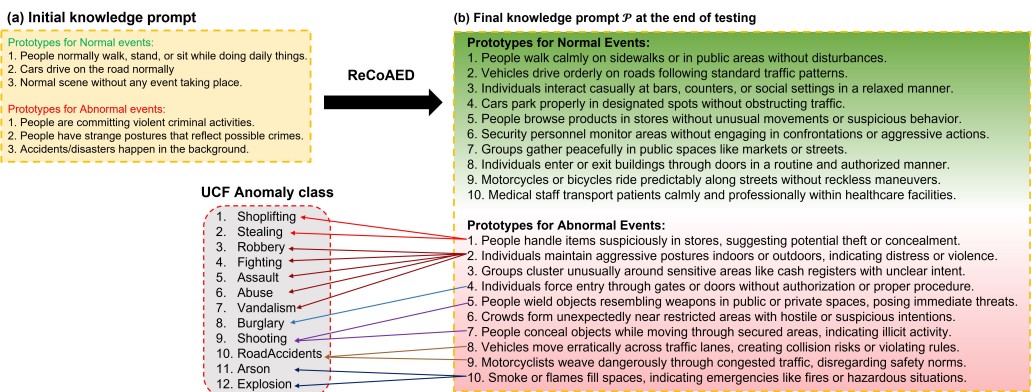

Figure 9: Visualization of the knowledge prompt $\mathcal{P}$ at the end of the testing on the UCF-Crime.

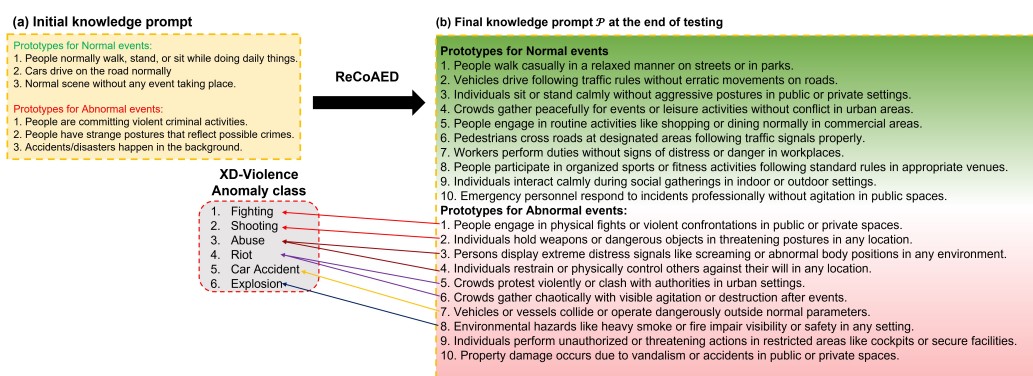

Figure 10: Visualization of the knowledge prompt $\mathcal{P}$ at the end of the testing on the XD-Violence.

It is important to note that the initial prompts did not contain any dataset-specific information or annotations regarding the types of abnormal events. This highlights the framework's strong capacity for self-evolution and knowledge discovery.

