# OpenReview forum: "Sparse Reasoning is Enough: Biological-Inspired Framework for Abnormal Event Detection with Large Pre-trained Models"
_ICLR.cc/2026/Conference — ICLR 2026 Conference Withdrawn Submission_

### Official Review · Reviewer_SYtd · 2025-10-29

**Soundness:** 3
**Presentation:** 3
**Contribution:** 2
**Rating:** 2
**Confidence:** 4

**Summary:**

While the paper proposes an interesting biological-inspired dual-stream framework (ReCoAED) for training-free Abnormal Event Detection (AED) and highlights the value of sparse reasoning, it suffers from critical gaps in methodological rigor, experimental comprehensiveness, analytical depth, and real-world applicability. These limitations undermine the validity, generalizability, and impact of the work, making it unsuitable for publication at ICLR 2026.

**Strengths:**

This paper proposes an interesting biological-inspired dual-stream framework (ReCoAED) for training-free Abnormal Event Detection and highlights the value of sparse reasoning.

**Weaknesses:**

1. Methodological ambiguities and incomplete design justification. The paper introduces a "dynamic memory M" to store representative frame records for the reflex stream, but its core management logic is underspecified, including no overflow or pruning strategy, and explains arbitrary threshold selection
2. The paper does not justify why these specific models were chosen over state-of-the-art alternatives (e.g., CLIP-ViT-L/14 for better feature quality, LLaVA-1.5 for VLM, or GPT-4o-mini for LLM). Without ablating model variants, it is impossible to determine if ReCoAED’s performance is tied to the framework itself or the selected models. The choice of core models (CLIP-ViT-B/16 for reflex, Qwen2.5-VL-7B for VLM, Deepseek-V3 for LLM) is arbitrary and unmotivated
3. The paper restricts experiments to two surveillance-focused datasets (UCF-Crime, XD-Violence), despite explicitly citing autonomous driving and industrial monitoring as key applications (Abstract, Section 1).
4. The paper reports small performance gains (e.g., +2.00% AUC over LAVAD on UCF-Crime) but provides no confidence intervals or t-tests. These gains could easily be due to randomness in frame sampling or test set shuffling.
5. The paper mentions "edge devices (e.g., camera, vehicle)" (Section 1) but only reports FPS on two Nvidia A100 GPUs (Table 3). It provides no data on the Memory footprint (critical for embedded GPUs like Jetson AGX Orin), or end-to-end latency (e.g., time to detect an anomaly in a 1080p video), or energy consumption (a key constraint for battery-powered edge devices).
6. The paper does not identify which abnormal events are not "enough" for sparse reasoning. For example, short-duration anomalies (e.g., a brief fight in a crowd) may be missed if the reflex stream filters the only frame containing the event. Also, gradual anomalies (e.g., a fire slowly spreading) may require dense sampling to track progression—does ReCoAED’s sparse approach delay detection?
7.  The paper references "structured instructions" for PVLM and PLLM (Section 3.4, Appendices C-D) but does not provide full prompt text.
8. The paper mentions "window smoothing" for reflex stream scores (Section 3.3) but does not specify the window size’s impact on temporal consistency.

**Questions:**

See the above comments

---

### Official Review · Reviewer_7eKs · 2025-10-31

**Soundness:** 2
**Presentation:** 1
**Contribution:** 2
**Rating:** 4
**Confidence:** 4

**Summary:**

This paper introduces ReCoAED, a biologically inspired framework for training-free abnormal event detection that mimics the human nervous system’s reflex and conscious reasoning processes to reduce redundant computation. ReCoAED consists of a lightweight reflex reacting stream that quickly filters and scores familiar frames using a CLIP-based feature–prompt matching mechanism and a conscious reasoning stream where a 7B-scale vision-language modelanalyzes only novel frames while a LLM periodically refines the event prototypes and memory, forming a closed feedback loop for self-improvement. Experiments on UCF-Crime and XD-Violence datasets show that ReCoAED achieves state-of-the-art training-free performance while using only 28.55% and 16.04% of the frames required by prior large-model methods, confirming that dense frame-level reasoning is unnecessary. Ablation studies demonstrate the effectiveness of its adaptive prompt updates, structured anomaly scoring, and temporal smoothing, while efficiency tests show over 100% speed improvement compared to LAVAD.

**Strengths:**

1. This paper introduces a unique, biologically inspired framework that simulates the human nervous system’s reflex and conscious reasoning processes, offering a fresh conceptual perspective for efficient abnormal event detection.
2. The paper includes detailed ablation studies, efficiency comparisons, and visualizations that clearly justify the role and effectiveness of each system component.
3. The proposed ReCoAED integrates a lightweight CLIP-based reflex reacting stream with a VLM–LLM conscious reasoning stream, forming an adaptive closed-loop system that balances speed and accuracy.
4. The method demonstrates that dense frame-by-frame reasoning is unnecessary—achieving state-of-the-art results while processing only 28.55% and 16.04% of frames compared to prior approaches, significantly reducing computational cost.

**Weaknesses:**

* The paper didn't follow the ICLR format. On page 7, the space after level heading does not satisfied the format requirements

1. The system depends heavily on particular large models. The claimed biological generality is undermined by the lack of tests with different architectures or scales, raising questions about robustness across model families.
2. Although computational savings are reported, there is no real-world or online deployment test. The framework’s latency, throughput under streaming input, and scalability to long or high-frame-rate videos remain unverified.
3. The multi-component loop involving CLIP, a large VLM, and an LLM introduces implementation and synchronization overhead. The coordination between the reflex and conscious streams, memory updates, and LLM reasoning cycles may be difficult to tune in practice.
4. While the paper emphasizes biological analogy, the interpretability of the reflex memory and the consistency of its decision boundaries are not deeply analyzed. The effects of prompt drift or LLM-generated prototype errors on long-term stability are unclear.
5. The paper provides an empirical demonstration that “sparse reasoning is enough,” but lacks theoretical justification or analytical evidence explaining why sparse reasoning suffices or under what conditions it might fail.

**Questions:**

- The LLM periodically updates the knowledge prompt. How do you prevent semantic drift or overfitting when the LLM rewrites prototypes? Is there any safeguard to maintain consistency over time?
- How well does ReCoAED generalize beyond surveillance datasets? such as industrial inspection, autonomous driving, or medical anomaly detection

---

### Official Review · Reviewer_eR3k · 2025-10-31

**Soundness:** 2
**Presentation:** 2
**Contribution:** 2
**Rating:** 2
**Confidence:** 5

**Summary:**

The paper solves Abnormal Event Detection (AED) and particularly focuses on developing a non-learned method by leveraging foundation models to analyze video frames. The paper largely builds on the recent work (Zanella et al. 2024) which uses a Vision-Language Model (VLM) to generate textual descriptions for each frame and then uses a Large Language Model (LMM) to analyze each description and report detected anomalies therein. The paper proposes to, instead of analyzing every single frame, sparsely analyze them as most frames are trivial that contain no abnormal events. The paper presents an algorithm called Reflex reacting and Conscious reasoning for AED (ReCoAED), which mimics human nervous system (Figure 1) to process familiar and unfamiliar signals. ReCoAED has multiple modules: (1) Reflex reacting module that measures similarity of a given frame with prototypes stored in a memory, intending to filter trivial frames, (2) Conscious reasoning module that generates event descriptions and anomaly scores. It also updates the prototype memory based on new anomaly detections. The paper uses the UCF-crime dataset to validate the effectiveness of the proposed method. It reports better AED results than previous methods by analyzing significantly fewer frames.

**Strengths:**

Below are notable strengths of this paper.
- Using image-based VLM to analyze video is interesting.
- The datasets used in experiments are standard in the literature.

**Weaknesses:**

Below are notable weaknesses of this paper.
- As the paper works on videos for AED, it is natural to question why not explore a video-based foundation model? Currently, it uses image-based VLM to analyze frames. But single frames do not have motion information of (abnormal) behaviors or events. The paper is expected to test video foundation models.

- Following the above, as the paper argues for not processing all the frames but better to sparsely analyze them, it is natural to ask whether using a video foundation model is more efficient? The reviewer is thinking about this paper "UAL-Bench: The First Comprehensive Unusual Activity Localization Benchmark" (WACV 2025) which uses a video foundation model to generate descriptions for video clips, and uses an LLM to analyze the descriptions for unusual activity localization.

- The paper uses the term "Abnormal Event Detection (AED)" while the closely related paper (Zanella et al. 2024) uses a different term "Video Anomaly Detection (VAD)". Yet, the two papers use the same datasets. It is unclear how the two tasks differ from each other? Why can the same datasets be used for two different tasks. Authors should clarify.

- Regardless of sparsely analyzing frames, the proposed method ReCoAED resembles the method LAVAD by (Zanella et al. 2024). However, the paper reports much higher numeric metrics of ReCoAED than LAVAD (Table 1 and 2). Can authors explain what helps ReCoAED? Does simply skipping frames for processing improve so much?

- From Figure 4 left, it seems hyperparameter "a" has a very small impact on the performance (i.e., within 1% AUC metric). Is AUC change in 1% statistically significant? If so, the paper is especially expected to discuss why the proposed ReCoAED can outperform the previous method LAVAD by 2% AUC (Table 1) (and 3.65% in AP).

**Questions:**

The reviewer asks the authors to address each point in weaknesses listed above and does not repeat them in this Questions box.

---

### Official Review · Reviewer_G9HJ · 2025-10-31

**Soundness:** 2
**Presentation:** 1
**Contribution:** 2
**Rating:** 2
**Confidence:** 3

**Summary:**

This paper introduces **ReCoAED**, a biologically-inspired dual-stream framework for training-free abnormal event detection (AED). The system leverages **lightweight CLIP-based filtering and VLM/LLM-based reasoning** to drastically reduce computation while maintaining or improving detection accuracy. Even though the authors claim that this algorithm is biologically inspired, the entire pipeline is somewhat incremental, and CLIP-based filtering for keyframe selection has already been utilized in recent video LLM papers. Additionally, the combination with advanced VLM (Qwen2.5-VL) and LLM (Deepseek) may be the reason for its improved performance.

**Strengths:**

1. Intuitively appealing biological analogy, including the reflex arc and conscious reasoning. Also, the feedback loop to refine prototypes and memory provides a clear self-evolving pipeline.
2. Experimental results demonstrate the effectiveness of this design with better performance and reduced runtime.
3. Following the recent trend in video anomaly detection (VLM+prompt in training-free setup).

**Weaknesses:**

1. The writing of this paper should be largely improved. While the biological analogy is motivating, the biological narrative doesn’t lead to formal insight or theoretical grounding.
2. The novelty is somewhat incremental: extending **LAVAD** and **AnomalyRuler** by adding adaptive frame filtering rather than introducing a fundamentally new detection principle. CLIP-based keyframe selection has already been investigated in recent video LLMs. Providing a comparison with keyframe sampling + VLM/LLM reasoning will be helpful.
3. The proposed method is only tested on advanced VLM + LLM (qwen2.5-vl and deepseek). What is the performance of LAVAD and AnomalyRuler on the same base pre-trained VLM/LLM?

**Questions:**

Please address my concerns in the weakness part, and I will consider increasing the score accordingly.

---

### Note · Authors · 2025-11-15

I have read and agree with the venue's withdrawal policy on behalf of myself and my co-authors.